

# Oral reading promotes predictive processing in Chinese sentence reading: eye movement evidence

Min Chang[1], Zhenying Pu[1] and Jingxin Wang[2,3]

[1] School of Education Science, Nantong University, Nantong, China
[2] Key Research Base of Humanities and Social Sciences of the Ministry of Education, Academy of Psychology and Behavior, Tianjin Normal University, Tianjin, China
[3] Faculty of Psychology, Tianjin Normal University, Tianjin, China

## ABSTRACT

**Background**. Fluent sentence reading is widely acknowledged to depend on top-down contextual prediction, wherein sentential and contextual cues guide the pre-activation of linguistic representations before encountering stimuli, facilitating subsequent comprehension. The Prediction-by-Production hypothesis posits an explanation for predictive processes in language comprehension, suggesting that prediction during comprehension involves processes associated with language production. However, there is a lack of eye movement evidence supporting this hypothesis within sentence reading contexts. Thus, we manipulated reading mode and word predictability to examine the influence of language production on predictive processing.

**Methods**. Participants engaged in silent or oral reading of sentences containing either high or low-predictable target words. Eye movements were recorded using the Eyelink1000 eye tracker.

**Results**. The findings revealed a higher skipping rate and shorter fixation times for high-predictable words compared to low-predictable ones, and for silent compared to oral reading. Notably, interactive effects were observed in the time measures (FFD, SFD, GD) during first-pass reading, indicating that word predictability effects were more pronounced during oral reading than silent reading.

**Discussion**. The observed pattern of results suggests that the activation of the production system enhances predictive processing during the early lexical access, providing empirical support for the Prediction-by-Production hypothesis in eye movement sentence reading situations, extending the current understanding of the timing and nature of predictions in reading comprehension.

# INTRODUCTION

Linguistic prediction involves the process by which listeners or readers anticipate upcoming words or linguistic structures based on contextual cues, stored prior knowledge, and language rules (*Corps, Gambi & Pickering, 2018*; *Huettig, 2015*; *Kuperberg & Jaeger, 2016*; *Pickering & Garrod, 2013*; *Ryskin & Nieuwland, 2023*; *Staub , 2015*). This predictive ability

Corresponding author
Jingxin Wang, wjxpsy@126.com

facilitates language comprehension by enabling the brain to prepare in advance for processing upcoming linguistic input, thereby enhancing the speed and efficiency of language comprehension (*Huettig, 2015*; *Ryskin & Nieuwland, 2023*). A substantial body of evidence demonstrates that prediction is engaged in reading, which justifies our focus on this area (for a review see *Staub , 2015*).

The empirical investigation of linguistic prediction often involves evaluating a word's cloze probability, which is determined through a sentence cloze task (*Taylor, 1953*). In this task, participants predict the next word in a sentence after being presented with its initial segment. Cloze probability presents the proportion of participants providing a particular word, offering a quantifiable measure of its predictability (*Staub , 2015*; *Taylor, 1953*). Empirical evidence spanning various languages, including Chinese, English, German, and Arabic, illustrates that word predictability significantly influences reading times, skipping rates, and refixation rates (*Aljassmi et al., 2022*; *Kliegl et al., 2004*; *Rayner et al., 2005*; *Rayner & Well, 1996*; *Wang et al., 2010*; for a review see *Staub , 2015*). Specifically, highly predictable words exhibit faster reading times and higher skipping rates than less predictable ones. Additionally, high predictability is associated with reduced N400 components (*Delong, Urbach & Kutas, 2005*; *Delong, Troyer & Kutas, 2014*; *Kuperberg & Jaeger, 2016*).

While extensive research has explored predictive processing in silent reading (for reviews see *Ryskin & Nieuwland, 2023*; *Staub , 2015*), our understanding of prediction in oral reading remains limited. Oral reading holds significant relevance, particularly for beginning readers (*Kragler, 1995*). Primary school children, for example, heavily rely on oral reading as a pivotal tool for augmenting text processing and facilitating memory. The additional articulation required in oral reading, as opposed to silent reading, allows individuals to process information at a slower pace, which facilitates deeper analysis and comprehension of complex materials, ultimately enhancing memory retention. This memory enhancement is supported by the "production effect," where the act of producing information—such as oral reading—leads to greater accuracy and durability of memory compared to passively receiving information, as in silent reading (*MacLeod et al., 2010*). These characteristics of oral reading may contribute to constructing contextual representations and retrieving related knowledge, thereby facilitating the preactivation of forthcoming information. Therefore, investigating predictive processing in oral reading is crucial, as understanding how this mode of reading engages predictive processing could offer valuable insights into the mechanisms of linguistic prediction.

The Prediction-by-Production hypothesis provides a theoretical framework for the predictive processing in language comprehension (*Huettig, 2015*; *Pickering & Gambi, 2018*; *Pickering & Garrod, 2007*; *Pickering & Garrod, 2013*). According to this hypothesis, language production and comprehension are interwoven, allowing individuals to predict both their own and others' utterances. This hypothesis suggests that comprehenders covertly simulate the linguistic form and actions of the speaker's expression, constructing a mental representation of the underlying communicative intention. Subsequently, comprehenders can then execute this intention through their production system to prepare for the

anticipated expression. This intricate process necessitates time, cognitive resources, and language production abilities (*Pickering & Gambi, 2018*).

Empirical studies have identified links between language production proficiency and predictive processing, providing support for this theoretical hypothesis (*Borovsky, Elman & Fernald, 2012*; *Drake & Corley, 2015*; *Hintz, Meyer & Huettig, 2016*; *Lelonkiewicz, Rabagliati & Pickering, 2021*; *Mani & Huettig, 2012*; *Martin, Branzi & Bar, 2018*). For instance, a recent study, by *Lelonkiewicz, Rabagliati & Pickering (2021)*, examined the impact of engaging the production system by introducing participants to read the sentence contexts either aloud or silently. The sentence contexts varied in the predictability of the final word. Results indicated that the contextual predictability effect was more pronounced when participants read the sentence contexts aloud compared to reading them silently (as demonstrated in their experiment 3).

Recent neurophysiological studies also provided compelling evidence for an intrinsic connection between language prediction in comprehension and production (*AbdulSabur et al., 2014*; *Gastaldon et al., 2020*; *Martin, Branzi & Bar, 2018*; *Silbert et al., 2014* for reviews see *Federmeier, 2007*; *Huettig, 2015*; *Ryskin & Nieuwland, 2023*). For instance, *Martin, Branzi & Bar (2018)* experimented by comparing three groups of participants tasked with reading highly structured Spanish sentences. These sentences contained either expected or unexpected noun phrases, serving as the primary task. The study measured lexical prediction effects using ERP (Event-Related Potential) N400 modulations on the article, comparing results across the three groups, each engaging in a different secondary task. The groups differed in the secondary tasks they performed while reading: syllable production (SP group), tongue-tapping (TT group), and syllable listening (SL group). The purpose was to assess the impact of taxing the production system on lexical prediction during sentence comprehension. The findings indicated that the expectation effect, observed as a reduced N400 response to expected relative to unexpected noun phrases, was diminished in the SP group compared to both the TT and SL groups. This suggests that when the production system is taxed (in this case, through the syllable production task, which likely prevents subvocal rehearsal of the verbal input), the ability to make predictions during sentence comprehension is hindered. *Martin, Branzi & Bar (2018)* thus offer the first evidence supporting the notion that prediction in reading comprehension is closely intertwined with language production.

For oral reading, there is a distinctive temporal and spatial gap, referred to as the eye-voice distance, that manifests between the visual input and pronunciation stages during reading. This eye-voice distance is temporally delimited to 500 ms and spatially spans 2–3 words (*Inhoff et al., 2011*). Readers are compelled to continually adapt fixation times on the target word or synchronize the eye-voice distance through regressive eye movements, thereby incurring elevated cognitive resource consumption for oral reading (*Ashby et al, 2012*). Moreover, the obligatory vocalization of each word during oral reading reduces the likelihood of skipping words, resulting in an augmentation of fixation numbers and duration, ultimately leading to a deceleration in reading speed (*Rayner, 2009*). In summation, the act of reading aloud is marked by a deliberate reduction in pace and an

augmented demand on cognitive resources, which could potentially alter the dynamics of predictive processing.

This leads us to hypothesize: does the increased cognitive demand of oral reading reduce the capacity for predictive processing? There are two possibilities. Given that oral reading needs more cognitive resources to coordinate the eye-voice distance to achieve a seamless information flow between predictive processing and bottom-up perceptual input, one possibility is that oral reading exhibits a lesser extent of predictive processing compared to silent reading, *i.e.*, diminished word predictability effect. Alternatively, according to the Prediction-by-Production hypothesis, which claims that language prediction could benefit from production, oral reading might show larger word predictability effects. Examination of the interplay between predictive processing and language production holds the potential to yield implications for our understanding of linguistic prediction. Thus, the current study aims to investigate the influence of reading mode on predictive processing.

Based on previous studies (*Lelonkiewicz, Rabagliati & Pickering, 2021*; *Martin, Branzi & Bar, 2018*), we adopted the typical sentence reading task, coupled with an eye tracker to record participants' eye movement behavior as previous pioneering studies did (*Rayner et al., 2005*; *Rayner & Well, 1996*). The sentence reading paradigm typically manipulates the predictability of target words inserted within a sentence. Participants read the sentence starting from its first word, mimicking a normal reading situation. Following *Liversedge, Paterson & Pickering (1998)*, early-stage eye movement measures, *e.g.*, skipping rate, first fixation duration (FFD), single fixation duration (SFD), and gaze duration (GD), are sensitive to predictive processing and always analyzed. Later eye movement measures, *e.g.*, regression path duration (RPD) and total reading time (TRT), indexing later semantic integration, are also analyzed.

Accordingly, in the present studies, we manipulated word predictability and conducted a comparative analysis of predictive processing during both silent and oral sentence reading. This approach aimed to explore how reading mode modulates predictive processing and to test whether the Prediction-by-Production hypothesis underlies prediction in reading. We anticipated the word predictability effect on early eye movement measures, hypothesizing that highly predictable words would exhibit shorter reading times (on FFD, SFD, GD, TRT, and RPD) and higher skipping rates than their less predictable counterparts. Additionally, we expected a robust reading mode effect, wherein oral reading would be slower than silent reading, characterized by increased fixation durations (on FFD, SFD, GD, TRT, and RPD) and a lower skipping rate.

Crucially, we hypothesized that if the Prediction-by-Production hypothesis underlies prediction in reading comprehension, we would expect interactive effects between reading mode and word predictability. Specifically, oral reading, which engages production processing, should produce larger predictability effects than silent reading on the early eye movement measures related to predictive processing (*e.g.*, skipping rate, FFD, SFD, and GD). Additionally, we propose that oral reading, by engaging language production mechanisms, may facilitate the generation of more precise word predictions compared to silent reading. The discrepancy between the anticipated and actual input words could potentially incur additional processing costs post-accessing the word representation

(termed Prediction Error Cost; *Frisson, Harvey & Staub, 2017*). Consequently, oral reading might engender a more pronounced word predictability effect on later-stage eye movement measures (*i.e.,* TRT and RPD), which are associated with semantic integration processes.

## METHOD

### Ethics approval

The research received approval from the research ethics committee at Nantong University (No. 50 in 2022) and was carried out under the principles outlined in the Declaration of Helsinki. All participants volunteered to participate in the experiment and signed informed consent.

### Participants

In the experimental design, which involves two random variables, *i.e.,* participant and item, we performed a statistical power analysis using the online tool PANGEA (*Westfall, 2015*). For detailed protocol, please refer to Supplement S1. According to *Brysbaert & Stevens (2018)*, a general effect size commonly employed in psychological research ($d = [0.3, 0.4]$) was utilized as the prior effect size for the analysis, double-tailed test, $\alpha = 0.05$. The power analysis indicated that to detect the anticipated interactive effect with 80% statistical power, a sample size ranging from 20 to 64 participants could be required. Thus we recruited 64 college students aged 17–26 years (mean = 20.6 years, 42 female) to participate in the formal reading experiment. All were native Chinese readers and naive to the purpose of the experiment. Each participant was paid 15 CNY (Chinese Yuan) after finishing the experiment.

### Materials and design

The experiment employed a within-subjects design, incorporating reading mode (aloud, silent) and word predictability (high, low) as independent variables.

For the assessment of word predictability, a cohort of 108 college students participated in the sentence cloze task. They were instructed to provide the next word based on the preceding sentence fragments truncated immediately before the target word. Subsequently, eighty-eight sentences were finally chosen, with the high and low predictable target words located in the same position in sentences, as shown in Table 1.

An independent sample $t$-test showed that the cloze value (*i.e.,* predictability) between high and low predictable words was significantly different (high: $M = 0.76$, $SD = 0.12$, range = [0.53, 0.95]; Low: $M = 0.03$, $SD = 0.03$, range = [0.009, 0.17]; $t\,(174) = 53.28$, $p < .001$). Moreover, to ensure comparability between the two conditions, we equated word frequency (*Cai & Brysbaert, 2010*; High: $M = 94.65$, $SD = 150.62$; Low: $M = 68.63$, $SD = 99.26$; $t\,(174) = 1.35$, $p = .178$), whole word complexity in stroke numbers (High: $M = 15.98$, $SD = 4.17$; Low: $M = 16.35$, $SD = 4.6$; $t\,(174) = 0.57$, $p = .572$), first character stroke number (High: $M = 7.68$, $SD = 2.5$; Low: $M = 8.16$, $SD = 2.91$; $t\,(174) = 1.17$, $p = .245$), second character stroke number (High: $M = 8.30$, $SD = 3.02$; Low: $M = 8.15$, $SD = 3.15$; $t\,(174) = 0.32$, $p = .751$), and sentence plausibility (High: $M = 3.98$, $SD = 0.31$; Low: $M = 3.91$, $SD = 0.27$; $t\,(174) = 1.53$, $p = .128$). These matching procedures

**Table 1** An example sentence of materials.

| Condition | Sentence | Target |
|---|---|---|
| High predictability | 本次开幕式邀请了各单位的领导进行开场致辞 | 领导 |
| low predictability | 本次开幕式邀请了各单位的代表进行开场致辞。 | 代表 |

Notes.

Target words are shown in bold. The high predictability word 领导 signifying "leader", contrasts with the low predictability word 代表 denoting "delegates". The sentence translates as "This opening ceremony invited leaders/delegates from various institutions to give opening speeches".

were employed to control potential confounding variables and ensure that any subsequent effects could be attributed to the manipulated factors of interest.

## Apparatus

An SR Eyelink 1000 eye tracker tracked right-eye movements during binocular viewing at a sample rate of 1,000 Hz. Stimuli were displayed in Song 26-point font as black text on a gray background (RGB: 192, 192, 192). The monitor was 19 inches and had a high resolution (1,280 × 1,024 pixels) with a refresh rate of 60 Hz. At 60 cm viewing distance, each character subtended 1° and so was of normal size for reading.

## Procedure

The experiment adopted a counter-balanced design consisting of four blocks using the Latin square method, as outlined in Supplement S2. Participants were randomly assigned to one of the counterbalanced lists. In blocks 1 and 2, participants initially read five practice sentences then the first half of the experimental sentences (44) and filler sentences (11) orally, followed by a rest screen directing them to read silently next. After this, they completed five additional practice sentences to adapt to the new reading mode, before proceeding with the second half of the experimental (44) and filler sentences (11). Conversely, for blocks 3 and 4, this sequence was reversed, *i.e.,* participants read the first half of sentences silently and then the second half of sentences orally. Notably, to facilitate participants' adaptation to the alternating reading modes, five practice sentences were presented at the outset and midway through the experiment. Even though the oral reading was not monitored by audio equipment, the participants completed the oral reading task with great seriousness and correctly. Each block encompassed 88 experimental, alongside 10 practice and 22 filler sentences.

The procedure was as previously described in *Chang et al. (2023)*; participants engaged in the study individually and were instructed to read normally and for comprehension. To ensure precise eye-tracking measurements, a 3-point horizontal calibration procedure was administered by the experimenter at the commencement of the experiment along the same line as each sentence presentation, guaranteeing spatial accuracy of .30° or better for all participants. Drift correction was checked before each trial and the eye-tracker recalibrated as required by the experimenter to maintain high spatial accuracy. At the initiation of each trial, a fixation square, equivalent in size to one character, appeared on the left side of the screen. Upon fixation on this location, the sentence was presented and participants pressed the space key once they finished reading the sentence. Subsequently, a comprehension

question, eliciting a yes/no response (with "yes" indicated by the "F" key and "no" by the "J" key), was presented following 27 sentences (16 questions following experimental sentences and 11 questions following filler sentences). For example, a sentence such as "此次画展的收入将捐献给一个慈善机构用于建设山区学校" (translated as "The proceeds from the exhibition will be donated to a charity to build mountain schools") was followed by a question like "此次画展的收入用于城市学校的建设吗?" (translated as "Will the proceeds from this exhibition be used for the construction of urban schools?"). The duration of the entire experiment for each participant was approximately 40 minutes.

## RESULTS

### Analysis

The participants exhibited a high level of accuracy in answering comprehension questions, with a mean accuracy of 92% ($SD = 3.8\%$, range = [82%, 100%]). Consistent with prior studies (*Lelonkiewicz, Rabagliati & Pickering, 2021*; *Martin, Branzi & Bar, 2018*), we analyzed to ascertain whether the two independent variables affect sentence comprehension. The regression analysis indicated a lack of statistically significant disparities in performance on comprehension questions across experimental conditions (reading mode: $b = -0.39$, $SE = 0.3$, $z = -1.28$; predictability: $b = -0.25$, $SE = 0.3$, $z = -0.82$; interaction effect: $b = -0.26$, $SE = 0.61$, $z = -0.43$). The nonsignificant difference in comprehension accuracy between oral and silent reading suggested that improved word prediction in oral reading is unlikely to be attributable to general attention or context integration factors that enhance comprehension accuracy. Following the exclusion procedures outlined in previous studies (*Aljassmi et al., 2022*; *Chang et al., 2020a*; *Chang et al., 2020b*; *Zhao et al., 2019*), fixations of short duration (<80 ms) and long duration (>1,200 ms) were excluded from the analysis. Additionally, trials characterized by tracking loss (14 trials), and head movement (20 trials) resulting from sneezing or coughing were excluded. Furthermore, trials associated with sentences receiving fewer than six fixations, impacting 131 trials, were also excluded. In aggregate, 160 trials (3.9%) were removed from the dataset (For detailed information on the data pre-processing procedures, please refer to the readme.docx).

Following *Chang et al. (2023)* and *Liversedge, Paterson & Pickering (1998)*, four eye-movement measures related to early lexical processing and two related to the later stage of semantic integration were analyzed. The early-stage measures include word-*skipping rate* (SKIP, probability of not fixating a word during first-pass reading), *first-fixation duration* (FFD, duration of the first fixation on a word during first-pass reading), *single-fixation duration* (SFD, duration of the first fixation on a word receiving only one first pass fixation), *gaze duration* (GD, sum of all first pass fixations on a word). The later-stage measures are *regression path duration* (RPD, the sum of all fixation durations beginning with the initial fixation on the target word and ending when the eyes exited the word to the right, including time spent rereading earlier words and time spent rereading the word itself) and *total reading time* (TRT, sum of all fixations on a target word).

The retained data underwent analysis using linear mixed-effects models (*Baayen, Davidson & Bates, 2008*) for continuous variables (FFD, SFD, GD, TRT, and RPD) and
generalized mixed-effects models for binomial variables (skipping rate). The analysis employed the lme4 package (*Bates et al., 2015*) in the R statistical environment (*R Development Core Team, 2016*). Maximum random-effects structures were incorporated into the models (*Barr et al., 2013*), with word predictability and reading mode as fixed factors, and participant and stimuli as crossed random effects. In instances where models failed to converge, the random-effects structure was trimmed, starting with adjustments related to stimuli. Reading times were log-transformed, although results for log-transformed and untransformed models exhibited similarity (see Supplement S3). Therefore, untransformed analyses are presented for transparency. By convention, $t/z$ values >1.96 were considered significant. However, given the adoption of six dependent variables, the probability of at least one false positive increases. Consequently, it is advisable to adopt a more stringent significance level of alpha = 0.01 ($|t/z|$>2.58).

## Result

Target word means were shown in Table 2, and statistical effects were summarized in Supplement S3.

### Word predictability effect

The observed word predictability effect was significant, manifesting in noteworthy impacts on various first-pass reading measures, including skipping rate ($b = -.18$, $SE = .07$, $z = -2.43$, $p < .05$), FFD ($b = 8$, $SE = 3$, $t = 2.48$, $p < .05$), SFD ($b = 9$, $SE = 3$, $t = 2.85$, $p < .01$), and GD ($b = 20$, $SE = 5$, $t = 4.18$, $p < .001$). High-predictable words exhibited shorter reading times and elevated skipping rates compared to their low-predictable counterparts, underscoring the robust nature of word predictability effects. Moreover, these effects extended to subsequent semantic integration measures, such as RPD ($b = 18$, $SE = 7$, $t = 2.55$, $p < .05$), and TRT ($b = 28$, $SE = 6$, $t = 4.85$ $p < .001$), revealing that readers invested more time in processing low-predictable words than their high-predictable counterparts.

### Reading mode effect

A distinct reading mode effect was identified, with main effects evident across both early and later measures($p$ s $< .001$)(skipping rate: $b = -1.92$, $SE = .08$, $z = -24.79$; FFD: $b = 62$, $SE = 5$, $t = 13.63$; SFD: $b = 71$, $SE = 5$, $t = 14.79$; GD: $b = 119$, $SE = 7$, $t = 16.72$; RPD: $b = 123$, $SE = 11$, $t = 11.12$; TRT: $b = 121$, $SE = 9$, $t = 13.69$). Specifically, silent reading exhibited higher skipping rates and shorter reading times compared to oral reading, indicative of a pronounced influence of reading mode on eye movement patterns.

### Interactive effect between reading mode and word predictability

Crucially, significant interactive effects emerged between reading mode and word predictability on some measures of first-pass reading, *i.e.,* FFD ($b = 18$, $SE = 6.12$, $t = 2.74$, $p < .01$), SFD ($b = 22$, $SE = 6.63$, $t = 3.35$, $p < .001$), GD ($b = 34$, $SE = 9.05$, $t = 3.8$, $p < .001$), as well as later measures RPD ($b = 34$, $SE = 14.26$, $t = 2.37$, $p < .05$) and TRT ($b = 23$, $SE = 11.02$, $t = 2.06$, $p < .05$), see Fig. 1. The interactive patterns were that the word predictability effects were significant during oral reading ($p$ s $< .001$)(FFD:

**Table 2  Means for target word measures.**

| Measures | Silent | | | Oral | | |
|---|---|---|---|---|---|---|
| | **High** | **Low** | **S_PE** | **High** | **Low** | **O_PE** |
| Skipping rate (%) | 44 (10) | 41 (10) | −3 | 12 (6) | 10 (5) | −2 |
| FFD (ms) | 219 (19) | 219 (18) | 0 | 277 (23) | 292 (25) | 15 |
| SFD (ms) | 219 (19) | 218 (18) | −1 | 282 (27) | 301 (29) | 19 |
| GD (ms) | 231 (22) | 233 (22) | 2 | 337 (34) | 372 (38) | 35 |
| TRT (ms) | 281 (37) | 298 (35) | 17 | 393 (42) | 431 (45) | 38 |
| RPD (ms) | 284 (51) | 286 (46) | 2 | 386 (49) | 421 (54) | 35 |

**Notes.**
Standard errors are in parentheses. The High and Low represent the high predictable and low predictable conditions, respectively. The Silent and Oral represent silent reading and oral reading, respectively.

FFD, first fixation duration; SFD, single fixation duration; GD, gaze duration; TRT, total reading time; RPD, regression path duration.

S_PE and O_PE represent the difference between High and Low predictable conditions when reading silently and orally, respectively.

[1]PE represents the mean difference between the high and low predictable conditions, as shown in Table 2.

PE[1] = 15 ms, $b = 16$, $SE = 3.84$, $t = 4.21$; SFD: PE = 19 ms, $b = 21$, $SE = 4.44$, $t = 4.69$; GD: PE = 35 ms, $b = 38$, $SE = 5.69$, $t = 6.64$, TRT: PE = 38 ms, $b = 40$, $SE = 7.37$, $t = 5.48$; RPD: PE = 35 ms, $b = 36$, $SE = 8.96$, $t = 4.01$). Conversely, these effects were not significant during silent reading (FFD: PE = 0 ms, $b = -1.3$, $SE = 4.82$, $t = -0.27$; SFD: PE = −1 ms, $b = -2.29$, $SE = 5$, $t = -0.46$; GD: PE = 2 ms, $b = 2$, $SE = 7.15$, $t = 0.27$; RPD: PE = 2 ms, $b = -0.45$, $SE = 11.26$, $t = -0.04$), except for TRT (PE = 17 ms, $b = 17$, $SE = 8.43$, $t = 2$, $p < .05$). Notably, the interactive effect did not reach significance for the skipping rate ($b = -0.09$, $SE = 0.15$, $z = -0.59$). When analyzing the data separately for silent and oral reading, the predictability effects on skipping rate were weak in both conditions (Silent reading: PE = 3%, $b = 0.14$, $SE = 0.08$, $z = 1.66$; Oral reading: PE = 2%, $b = 0.23$, $SE = 0.12$, $z = 1.81$).

### *Influence of reading speed on predictive processing*
To investigate whether the observed enhancements in predictability effects on reading times were primarily driven by the slowdown associated with oral reading, we conducted a post-hoc analysis (potentially underpowered to detect complex interactions) examining the moderating effect of reading speed. Subjects were divided into fast and slow reading speed groups based on the median reading speed within each block. This grouping variable was incorporated as a factor in the statistical model. Notably, this categorization process was carried out separately for oral and silent reading modes. The results of this analysis are detailed in Supplement S4. The analysis revealed no significant two-way interactions between reading speed group and word predictability across all measures ($|t/z|s < 1.03$), indicating that reading speed group did not modulate the effect of word predictability. Furthermore, no significant three-way interactions were observed across all measures ($|t/z|s < 0.94$). However, the reading mode exhibited a robust modulating effect on the word predictability effects on the early eye movement measures (FFD: $b = 16$, $SE = 6$, $t = 2.55$, $p < .05$; SFD: $b = 21$, $SE = 7$, $t = 3.13$, $p < .01$; GD: $b = 34$, $SE = 9$, $t = 3.75$, $p < .001$) and wearker modulation on measures concerning later semantic integration (TRT: $b = 21$,

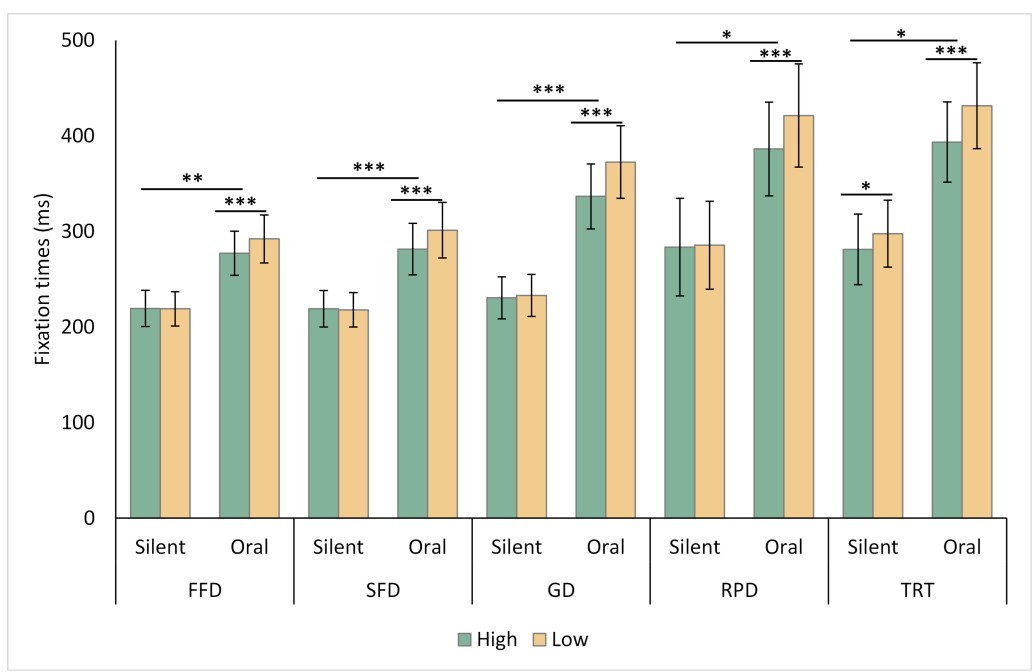

**Figure 1** The interactive pattern between reading mode and word predictability. Note: * $p < .05$, ** $p < .001$. *** $p < .001$. FFD, first fixation duration; SFD, single fixation duration; GD, gaze duration; TRT, total reading time; RPD, regression path duration. Error bars represent standard error.

$SE = 11$, $t = 1.91$, $p = .056$; RPD: $b = 34$, $SE = 14$, $t = 2.32$, $p < .05$), suggesting that the reading speed group does not modulate the interaction between reading mode and word predictability.

To further explore this relationship, we also performed an ANCOVA analysis using reading speed as a continuous variable. The results (as presented in Supplement S5) confirmed that the two-way interactions between reading mode and word predictability remained significant for FFD, SFD, and GD ($|t|$ s > 2.77, $p$ s < .01), TRT, and RPD ($|t|$ s > 2.05, $p$ s < .05), even when controlling for reading speed.

Overall, the findings reveal highly significant interactions between reading mode and word predictability on the early eye movement measures (FFD, SFD, and GD, $p$ s < .01) and significant interactions on measures related to later semantic integration (RPD and TRT, $p$ s < .05). However, caution is warranted in interpreting the interactions for RPD and TRT due to the potential for false positives. In addition, the post-hoc analyses suggest that variations in reading speed did not significantly affect the interactions between reading mode and word predictability effects, particularly on the early eye movement measures (FFD, SFD, and GD). Therefore, these statistical results strongly support the conclusion that reading mode modulates word predictability effects on early eye movement measures that index lexical access.

## DISCUSSION

In this study, we investigated whether engaging in oral reading, which entails the language production process, heightens predictive processing in Chinese sentence reading. Participants read sentences containing target words with either high or low predictability levels, either silently or aloud. The results unveiled substantial effects of reading mode and word predictability across some early and late eye movement measures. Particularly noteworthy were the interactive effects observed in some of the first-pass eye movement measures (FFD, SFD, GD), where word predictability effects proved more pronounced during oral reading compared to silent reading. These findings provide support for the Prediction-by-Production hypothesis in the context of sentence reading.

In line with findings from previous research (*Aljassmi et al., 2022*; *Chang et al., 2020a*; *Chang et al., 2020b*; *Cui et al., 2022*; *Kliegl et al., 2004*; *Liu et al., 2018*; *Rayner & Well, 1996*; *Wang et al., 2010*; *Zhang et al., 2020*; *Zhao et al., 2019*), this study identified significant effects of word predictability on both early (skipping rate, FFD, SFD, and GD) and late eye movement measures (RPD and TRT), elucidating that readers' pre-processing of high-predictable words affects early word access and later semantic integration between the word and sentences. However, as illustrated in Table 2, while our results replicate some aspects of the predictability effects observed in previous studies, these effects appear to be specific to the context of oral reading and are not as evident in silent reading. One possible explanation for this discrepancy might be that the cognitive and motor demands involved in oral reading may amplify the effects of predictability. When reading aloud, individuals engage in two simultaneous tasks: decoding the text (*i.e.,* processing and understanding the words) and producing speech (*i.e.,* vocalizing the words). The interaction between these tasks and the predictability of the word may lead to different outcomes compared to silent reading, where only decoding is involved. Relatively speaking, the task of silent reading may be less demanding, allowing both high- and low-predictable words to be processed quickly. As a result, the predictability effect in silent reading might not be as pronounced.

Furthermore, the present study found the pivotal role of reading mode in shaping the dynamics of eye movements during the reading process. Notably, it reveals distinctive patterns, such as a slower reading pace and increased fixation times, when readers engage in the act of oral reading. The most obvious requirement of oral reading is that each word must be pronounced. Thus only a few words will be skipped, as found in previous studies (*Ashby et al, 2012*; *Coupé et al., 2014*; *Gao et al., 2016*; *Vorstius, Radach & Lonigan, 2014*), which slow reading speed. In addition, oral reading is characterized by a reduced perceptual span and diminished parafoveal preview processing compared to silent reading, which can affect the reader's ability to preprocess adjacent text (*Ashby et al, 2012*; *Zhang et al., 2020*). Additionally, oral reading typically involves higher refixation rates and prolonged fixation durations, factors that collectively contribute to a slower reading speed. Moreover, effective oral reading demands meticulous coordination between eyes and voice. This coordination is crucial for maintaining optimal eye-voice distance and for maximizing reading efficiency. In the context of oral reading, therefore, readers were in an enhanced cognitive processing

situation, necessitating sustained attention and heightened utilization of cognitive resources for tasks such as pronunciation control and self-monitoring.

Most importantly, the present study revealed that reading mode significantly influences word predictive processing during the first-pass reading. The effect of word predictability was more pronounced FFD, SFD, and GD during oral reading compared to silent reading, indicating that oral reading facilitates predictive processing. This finding is in line with the "Prediction-by-Production" hypothesis (*Pickering & Gambi, 2018*). According to this hypothesis, individuals employ the language production system to anticipate linguistic outcomes, and the active utilization of this system significantly enhances predictive capabilities. Evidence from previous studies supports this notion (*Drake & Corley, 2015*; *Hintz, Meyer & Huettig, 2016*; *Ito & Pickering, 2017*; *Lelonkiewicz, Rabagliati & Pickering, 2021*; *Martin, Branzi & Bar, 2018*; *Pickering & Gambi, 2018*; *Pickering & Garrod, 2007*; *Pickering & Garrod, 2013*). Our current study, embedded within the context of the sentence reading paradigm, provides further reinforcement for the idea that language production plays a pivotal role in language prediction, thereby strengthening the foundations of the "Prediction-by-Production" hypothesis.

However, the interaction between reading mode and word predictability was not observed for skipping rate, a measure that reflects the earliest parafoveal processing. According to the prediction-by-production hypothesis, oral reading engages the language production system, which should theoretically enhance predictive processing. Therefore, we expected to observe a more pronounced predictability effect on measures associated with predictive processing, including skipping rate, first fixation duration (FFD), single fixation duration (SFD), and gaze duration (GD). The absence of a significant interaction in the skipping rate is inconsistent with these expectations. One possible explanation for this null interaction is that the effect of word predictability on skipping rate in Chinese reading may not be particularly robust. While previous studies have reported significant predictability effects during silent reading (*e.g.*, *Rayner et al., 2005*; *Zhao et al., 2019*), other research has found no significant effects (*e.g.*, *Chang et al., 2020a*; *Cui et al., 2022*; *Zhao et al., 2021*) on skipping rate. In the present study, when we analyzed the data separately for silent and oral reading, we found that the effects of word predictability were relatively weak in both conditions. Furthermore, the word-by-word reading pattern required in oral reading may significantly reduce the likelihood of skipping words in high (12%) and low (10%) predictability conditions, potentially leading to a floor effect. This contrasts with the expected larger predictability effect in oral reading.

The more pronounced predictability effect observed during oral reading suggests that the heightened cognitive load associated with oral reading did not negatively impact predictive processing. Notably, our study participants were college students, a demographic known to generally perceive oral reading as manageable and to exhibit peak cognitive capacities throughout the lifespan. This cohort demonstrated more pronounced predictability effects during oral reading, potentially owing to their heightened cognitive abilities. However, it is important to consider that the prediction-by-production may be a resource-intensive mechanism (*Pickering & Gambi, 2018*). Consequently, comprehenders with varying reading abilities (*e.g.*, non-native speakers, individuals with low literacy levels,

children, and older adults), may rely on this mechanism less frequently than typical native young adults. This warrants further exploration in future research.

In addition, the pace of reading might influence predictive processing in language comprehension. When readers opt for oral reading, their reading speed tends to slow down. This slowing enables individuals to allocate more time to process the intricate syntax and semantic information within sentences, thereby enhancing their overall understanding (*Kragler, 1995*). The extended processing time facilitates the formation of accurate mental representations of the contextual information from preceding text and the extraction of pertinent details from long-term memory, thereby facilitating the prediction process. To test this guess and further consolidate the role of the production system in promoting predictive processing, we conducted additional analyses. Specifically, we introduced a group factor based on reading speed to explore how variations in reading speed may influence predictive processing. The results of these analyses revealed a non-significant two-way interaction between the group factor and predictability effect, as well as a non-significant three-way interaction between the group factor, reading mode, and word predictability. These findings suggest that differences in reading speed did not substantively account for the observed enhancements in predictability effects on early eye movement measures during oral reading.

However, it is important to note that this post-hoc analyses may not fully disentangle the confounding effects of reading speed, as slower reading is an intrinsic characteristic of natural oral reading. One potential method to better isolate the effects of language production processes and reading speed on contextual predictability would be to utilize a rapid serial visual presentation (RSVP) paradigm. This would allow participants to read sentences either silently or orally at a controlled speed, minimizing the confounding influence of reading speed. Additionally, we could examine how readers' language production abilities affect predictive processing in both reading modes under the RSVP paradigm. Manipulating stimulus onset asynchrony (SOA) could also provide further insights into the relationship between reading speed and contextual predictability. These ideas are preliminary and would require further refinement and validation, but we believe they represent a promising direction for future research.

Regarding the interaction effects between reading mode and word predictability on RPD and TRT ($p$ s < .05), we are hesitant to accept these findings due to our preference for a more stringent significance level of alpha = .01 to reduce the likelihood of false positives. Should these interactions exist, the underlying mechanism may be elucidated by a previous study employing event-related potentials (ERPs), which have highlighted the influence of SOA (Stimulus Onset Asynchrony) conditions on the levels that are pre-activated (*e.g.*, semantic, phonological, and word-form levels). For instance, *Ito et al. (2016)* observed that under slower SOA conditions, words orthographically related to the predictable word elicit smaller N400 components. This finding implies that, at a slower reading speed, readers can form more specific word form predictions. In contrast, under faster reading conditions, a probabilistic semantic prediction may take precedence (*Ito & Pickering, 2017*). The additional temporal resource afforded by a slower reading pace, enables readers to formulate more robust and certain predictions, activating information

at the specific word form level. Consequently, a discrepancy between the actual input and anticipated words, after accessing the word representation, incurs additional processing costs (*i.e.,* Prediction Error Cost) post-accessing the word representation (*Frisson, Harvey & Staub, 2017*; *Freunberger & Roehm, 2017*). Thus, readers need to allocate more time to low predictable words on RPD and TRT to reconcile prediction errors when discrepancies happen.

In summary, this study tentatively explores the influence of language production on word predictability effects in sentence reading, leveraging eye movement measures as behavioral indicators. We found that the reading mode modulates the word predictive processing, with a larger word predictability effect for oral than silent reading. The current results, from the sentence reading context, provide further support for the "Prediction-by-Production" hypothesis.

### Future directions

Previous research, alongside the current study, has predominantly conjectured the association between language production and prediction from a behavioral perspective. However, a notable lack exists in the exploration of the neurophysiological underpinnings connecting production and prediction in these investigations. Looking ahead, it is imperative to employ more sophisticated psycholinguistic paradigms to gain a deeper understanding of whether prediction and production are rooted in similar neurophysiological mechanisms. Related exploration could provide more evidence for the prediction-by-production hypothesis and help in developing production-based therapy for reading disorders.

In addition, our study implemented a strictly controlled experiment design where only one sentence was displayed on the screen at a time. While this approach allows for precise measurements, it may limit the generalizability of our findings to more naturalistic reading situations. Recent trends are moving towards more ecologically valid experimental designs. For instance, research by *Carter et al. (2019)* and *Shain et al. (2020)* employs cohesive story-based stimuli that integrate analyses across different linguistic levels (*e.g.*, word, sentence, discourse). Such methodologies improve the ecological validity of language research. Future research needs a strictly controlled experimental design, and attention should also be paid to improving the ecological validity of the study.

## CONCLUSION

The present study investigated predictive processing during both silent and oral reading, revealing a more pronounced predictability effect in the context of oral reading. This finding implies that the activation of the production system enhances predictive processing during the early lexical access. Consequently, our study offers empirical support for the Prediction-by-Production hypothesis, extending the current understanding of the timing and nature of predictions in reading comprehension.

### Funding
The research was supported by a grant from the National Natural Science Foundation of China to Jingxin Wang (No. 32271119), the Key Project of Educational Science Planning Programs of Jiangsu Province to Min Chang (B/2023/01/109), and the Large Instrument Open Fund of Nantong University to Min Chang (project number: KFJN2205 Changmin). The funders had no role in study design, data collection and analysis, decision to publish, or preparation of the manuscript.

### Grant Disclosures
The following grant information was disclosed by the authors:
National Natural Science Foundation of China to Jingxin Wang: 32271119.
Key Project of Educational Science Planning Programs of Jiangsu Province to Min Chang: B/2023/01/109.
Large Instrument Open Fund of Nantong University to Min Chang (project number: KFJN2205 Changmin: KFJN2205.

### Competing Interests
The authors declare there are no competing interests.

### Author Contributions
- Min Chang conceived and designed the experiments, performed the experiments, analyzed the data, prepared figures and/or tables, and approved the final draft.
- Zhenying Pu performed the experiments, prepared figures and/or tables, and approved the final draft.
- Jingxin Wang conceived and designed the experiments, authored or reviewed drafts of the article, and approved the final draft.

### Human Ethics
The following information was supplied relating to ethical approvals (i.e., approving body and any reference numbers):
The Nantong University granted Ethical approval to carry out the study within its facilities (No. 50 in 2022).

### Data Availability
The raw data is available at figshare: Chang, Min (2024). Raw data and data for analysis in Rstudio. figshare. Dataset. https://doi.org/10.6084/m9.figshare.25054937.v6

### Supplemental Information
Supplemental information for this article can be found online at http://dx.doi.org/10.7717/peerj.18307#supplemental-information.

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
