# Peer review of "Oral reading promotes predictive processing in Chinese sentence reading: eye movement evidence"

_PeerJ, doi:10.7717/peerj.18307_

## Round 0.1 · original submission · Major Revisions

Dear Authors,

Both reviewers have highlighted some major concerns at the theoretical and methodological levels. After reviewing the manuscript myself, I fully agree with the Reviewers. Therefore, I urge you to carefully consider each point raised by the reviewers and address all issues before resubmission.

At the theoretical level, I suggest clarifying the extent to which the reported results relate to prediction-by-production models. As suggested by one of the Reviewers, analyzing the comprehension questions might aid in interpreting the results. I recommend considering justifying of the multiplicity of dependent variables considered and addressing the issue of multiple statistical testing. Furthermore, it is important to better describe how you transformed the data before statistical analyses and to fully report the models used. Lastly, but very relevant, there appears to be an ethical issue concerning the consent given by participants for data sharing.

Please keep in mind the following caveat as you revise the paper: Inviting resubmission does not guarantee that the next version, or any subsequent version, will be accepted for publication. Moreover, the clarifications resulting from the revision may reveal new issues that were previously unnoticed, which could preclude publication. I am not pre-judging this particular case; this is simply a precautionary note I provide to all authors before inviting resubmission.

·

Basic reporting

- The paper does not explicitly define linguistic prediction, and does not address any alternative or competing theoretical models of predictive language processing beyond prediction-by-production. Because the findings are, in my interpretation, not logically conclusive of prediction-by-production, I think the manuscript could benefit from addressing alternative theoretical views.

- The discussion section contains a few paragraphs that I think should be moved to the introduction, because they exclusively discuss previous studies without reference to the current findings: l 294-313 (which almost literally repeats information from the introduction), l 314-325, l 333-343.

- The paper is generally written in clear and correct English, but several formulations stand out. A few examples: “anticipatory predictions” (l 27) -> predictions are by definition anticipatory; “minor N400 components” (l35) -> suggestion: “reduced N400 components”; “facilitating memory improvement” (l 46); “The predictability between high and low predictable words was significant” (l 163); “readers were immersed in an augmented cognitive processing realm” (l 286) all need rephrasing.

- Tables and figures are generally clear. A few suggestions: provide a legend for the abbreviations in Tables 3 and 4 in a table note; same for Figure 1; “The Interactive Effects” is not an informative title for Figure 1 -> please describe this in terms of the variables involved.

Experimental design

- The materials for this study are well-designed. There is sufficient statistical power and the counterbalancing design is appropriate. Operationalization of reading time measures and data cleaning are all fairly standard. However, information is lacking about the oral reading procedure: what instructions were provided, how (if at all) was compliance monitored, performance (e.g., speed, errors) measured, et cetera?

Validity of the findings

- I don’t think this design allows one to differentiate between engagement of the production system per se and the general slow-down that comes with oral reading as the main driver of enhanced predictability effects on reading times. The authors do mention this latter possibility, but frame it as an additional, rather than alternative explanation (l 326). I do not see how the evidence warrants this particular framing. The evidence seems equally consistent with enhanced engagement of production mechanisms, slow-down (which might provide the time for those very prediction-by-production processes to run their course, but could similarly provide the time for other, relatively slow combinatorial prediction mechanisms that do no necessarily involve the production system), and a combination of these. (See, for example, Huettig & Guerra 2019 and Wlotko & Federmeier 2015 on the effects of timing on prediction.) That does not preclude this study from being an interesting first step, but I think this caveat should be more extensively discussed in the final sections of the paper.

- At the same time, oral reading requires “sustained attention and heightened utilization of cognitive resources for tasks such as pronunciation control and self-monitoring” (l 286-287). Given that the availability of cognitive resources is a major determinant of whether prediction-by-production occurs (see Pickering and Gambi 2018), this property of oral reading would seem to lessen the probability of prediction while reading aloud. I think that possibility warrants further discussion – perhaps even in the introduction.

- One of the main selling points (as framed by the authors) for this study is that it provides a test for the prediction-by-production model in a natural reading context. However, I don’t see what is particularly natural about reading a large number of isolated sentences from a screen – and half of them aloud. This may be a less contrived task than some of the studies cited, but calling it “natural” seems a bit of a stretch. I think the authors could make this argument with more precision and nuance. In particular, I would like to see how the authors see their study as more natural than Lelonkewiecz et al. 2021, to which it has some general similarities.

- Word skipping effects (which take up a considerable part of the results section) are conspicuously absent from the discussion section. More generally, I think the authors can be more precise and thorough in characterizing the oral reading process (both in their study and in the general literature) than stating that “it reveals distinctive patterns, such as a slower reading pace and increased fixation times” (l 282-283).

Additional comments

- Overall, I think this is an interesting paper that holds a potentially valuable contribution to the literature. I have little to argue with the method and data analyses, but I think the contextualization of the study and the interpretation of the findings need more depth and precision before it is ready for publication.

·

Basic reporting

This study tested the prediction-by-production hypothesis by investigating the effect of reading aloud vs silently on sentential prediction. To capture prediction, several eye-tracking measures were taken. Overall, I think this paper has the potential of contributing important evidence and, ultimately, I would hope to see it published. However, in its present form the manuscript does not contain the information necessary to properly evaluate the quality of the study. I also have some other concerns that I outline below.
Literature review and study motivation –
lines 26-28; P & G 2013 theorise about language comprehension in general, not only reading
lines 35-27; I sincerely hope high predictability is not related to ‘reduced brain activation’... please be more specific.
Lines 44-56; I find the structure of your argument a bit confusing – you first seem to argue reading aloud might be helpful (e.g., in dyslexia), but then from line 50 you mention it is also associated with increased cognitive load. So what predictions did you make – that reading aloud vs silently will improve or impair word comprehension?
Lines 55-56; you claim your work is important for reading difficulties, but you never show how/why.. either drop this claim or justify please.
Line 65; ‘deliberate reduction’ implies people slow down intentionally; cite evidence if this is the case.
Line 69 and onwards; there is no single prediction-by-production model; if you want to generally refer to the idea, I suggest you call it a hypothesis; otherwise be precise which model you refer to at each time (P & G 2013 or another one)
Line 73; assuming you refer to P & G 2013, I don’t think comprehenders mimic only the form.. consider rephrasing
Line 77; what language abilities
Lines 80 and elsewhere in the intro; Ruth Corps did some interesting work on prediction; you also forgot to cite the study by Clara Martin which shows production effects for the first time
Line 103; you argue your experiment is novel because it examined prediction-by-production in a natural reading context; I disagree – your study is an (ingenious) combination of classic eye-tracking studies like those of Rayner and the behavioural study of Lelonkiewicz et al., 2021; your participants did not read text passages, they did the very standard lab task of viewing isolated sentences while being eye-tracked and with calibration breaks between trials etc. This is far from natural reading. I suggest you drop this argument. I also think you should discuss Rayner and Lelonkiewicz studies in the introduction as they are very close to what you did.
On the topic of discussing prior work, when referring to past research on prediction, please cite also: Federmeier 2007 Psychophysiology, and the recent Ryskin & Niuewland 2023 TICS; the 2015 Heuttig in Brain Res is also a very good review
Lines 110-115; are you referring to prediction error and its possible impact on word processing? This is interesting – did this inspire any of your predictions for this study? In general, it is not clear to me what were your predictions for the different eye-tracking measures and the reasons for using that many. I return to this point in the experimental design section.
Language – please go carefully through the ms, ideally with a fluent English speaker. Overall the language is correct, but occasionally I did not quite understand some of the expressions you used (I give examples throughout this review). There are also some minor mistakes such as using singular ‘stimulus’ instead of plural ‘stimuli’ etc.
Raw data – the authors submitted a summary spreadsheet and another more complete spreadsheet with the data. Actual raw (.edf) files were not submitted. In my opinion, this is fine - .edf files are large and less accessible than excel – but I flag this since the editorial criteria mention raw data should be attached.
Figure 1 – you should show data from all RT measures. Please make sure the caption provides the following details: full names of the measures shown (now there are only abbreviations, e.g., FFD); what does * signify? What are the error bars? You say engaging production ‘amplifies the impact of word predictability on eye movement measure’ – this sounds somewhat confusing; you used eye-tracking to measure the predictability effect, didn’t you? Consider rephrasing. Consider anchoring the y axis at 0 as this is the baseline value of your DVs.
Table 2 – what do you mean by ‘norming data’? Report degrees of freedom, type of t-test, hypothesis type.
Table 3 – make sure you call your experimental conditions the same way throughout the ms. For example, there are different labels for predictability levels in table 2 vs 3, or reading mode levels in table 3 vs figure 1.
Table 4 – my personal preference would be to have results from the complete model structure in the supplement and report the critical main and interaction effects in text. I do not think you need this table here. This is just a suggestion though.

Experimental design

My main concern wtr the experimental design is about your eye-tracking measures. In total, you measured and tested for the predictability effect 6 times (5 RT measures and 1 skipping). It is not clear to me why did you include all – if you expected the same pattern to emerge then why test all? Please explain, ideally referring to past studies that employed these measures to look for predictability effects. I also think you should consider addressing the issue of multiple comparisons (inflated false positives) in your statistical analysis – this could be done by applying p-value corrections or a MANOVA like model, for example.
Second, please submit a complete analysis code that uses the spreadsheet with more variables (not the summary one). This way, it will become possible to understand how did you extract and calculate your variables, transform data, compute means, exclude participants etc. It will also become clear what was the final structure of your models. For the sake of transparency/reproducibility, please share the full script. For the sake of explaining your study to the reader, please make sure your methods section gives a complete description of all the analysis steps and the models you have run.
Third critical issue concerns ethics. According to the English translation of the consent form, you declare that no data will be ‘given to others’ without participant’s permission; yet, you shared data from the experiment with PeerJ (and you might want to make them publicly available upon publication using platforms like OSF); if the English translation is faithful to the Chinese original, then you are prohibited from sharing data unless you seek permission from all your participants; please make sure to do so prior to the next round of review. On the point of privacy, what are the participant labels in the ‘pp’ column of the dataset? How were these labels assigned to individual participants?
Line 131; participants ‘volunteered’ meaning they gave consent, or that they came to you and asked if you are running any studies they could take part in?
Line 133-142; could you please give the standard details about the sample, e.g., age, gender, sight, reading/learning disorders? How did you recruit and reimburse them? Re power analysis, it is what it is (next time you can run simulations of your LME in R; consider also pre-registering), but please describe the structure of the model you used to run the sample estimation and share the protocol from the shinyapp if possible.
Lines 162-173; I am not familiar with the literature on reading in Chinese, for other reader like me, could you explain why you controlled for whole/first/second stroke numbers? Are these measures of visual complexity?
Line 179; ‘refresh rate’
Line 181: ‘procedure’
Lines 182-198; please spell out what IVs were manipulated within/between and what did it mean in practical terms (e.g., half of the participants first read aloud and then.. etc); ‘recalibrated as required’ by what or whom, please explain; comprehension questions were presented at 20% trials, but 0.2*88 = 17,6 questions; please give an example of the questions; the duration of the experiment was 25 minutes, but in the consent form you invite people to take part in a 10 minute experiments; I do not think this is necessarily a major problem, but please see my earlier point about participant payment.

Validity of the findings

The multiple comparison and model description points I outlined in the previous section need to be addressed before I can commit to evaluating your findings. Here, I list other issues:
Lines 201-207; please report if there were between conditions differences in comprehension questions – this has to do with the possibility that better word prediction in reading aloud could be due to general attention or context integration etc; what ‘standard procedures’ for RT exclusion? To my knowledge, there are many - please elaborate (and provide the complete script where you carry out RT trimming). ‘maximum random-effect structures were incorporated’ – I am afraid this is not the case at least in the R script you provided now; please say what versions of R and R packages you used
Lines 236-246; report interaction terms when making claims about interaction effects. Although it is also useful to report condition means (with tests for differences between those means if you like)
Lines 247-260; for the GLMER on skipping, you compare model with vs without interaction (how did you compare the models? Say if you used anova function). This is a perfectly valid approach, but I do not understand why did you do it only for this model – if you trust the significance testing of your interaction terms, then why bother doing this extra analysis? If you don’t trust them, then why did you not do this analysis for all models?
As for the discussion, consider moving the discussion of Lelonkiewicz et al to intro. Please tone down the claim that you used natural reading (see my comments above).
Could you elaborate what exactly is your contribution to the theory of prediction by production? For example, some of the studies you cite make claims about what/when/how utterances are predicted (Huettig, 2015) or support one theoretical model over another. How does your study contribute, except for using eye-tracking over ERPs or behavioural measures.
Line 277-278; explain why your data suggest predictability ‘affects early word access and later semantic integration’
Lines 285-287; I don’t really know what you mean by ‘augmented cognitive processing realm’, could you please elaborate or rephrase
Line 317; cite the studies who observed these findings
Line 324-325; which model you refer to?
Line 328; ‘deliberate’ slowing? See my point above; also, data from the comprehension questions will help us understand if reading aloud was associated with deeper processing; Martin et al and Lelonkiewicz et al did test for differences in overall comprehension, you might want to see what they found
Line 355; and what would we learn from seeing that prediction and production are related at the neuro level? I am asking this question again to encourage you to think about theory and how you could contribute to it

Additional comments

Full R script and other details pending, I think your study is valuable and contributes important data. I wish you best of luck with your revision!

---

## Round 0.2 · Major Revisions

Dear Dr. Chang,

I was fortunate to receive the comments from both reviewers who reviewed the manuscript in the first round. As you will see, both acknowledge the substantial improvement in the new version. However, they believe that more work is still required on the manuscript. In particular, Reviewer 2 lists a series of important issues.

I fully agree with the comments of both reviewers and recommend carefully considering them before submitting a new version of the manuscript.

- Furthermore, while reading the manuscript myself, I found additional issues that I would like you to consider.

- In lines 66-70, you seem to suggest that oral reading is crucial for people with dyslexia as a compensatory strategy for improving comprehension. This is not consistent with existing literature (including the one cited, Pedersen et al., 2016), which shows that reading aloud (as opposed to silent reading) is associated with more difficulties in comprehension. Please clarify this point.

- In the “Materials” section, it would be helpful to provide the cloze probability value for each sentence as well as its English translation.

- In the paragraph “The influence of reading speed on predictive processing,” it is not clear how the median reading speed was computed. I think it would not be appropriate if the median value were computed by pooling oral and silent reading together. Fast silent readers might not be fast oral readers, and vice versa. Additionally, I do not understand how the results of this analysis can exclude the hypothesis that reading speed might be the basis for the predictability differences observed in the experiment.

Finally, regarding the consent to data sharing, I double-checked with the Editorial Office and we agreed that the new version of the consent form resolved the issue. Items 4 and 5 of the consent supersede the statement, "All data and information collected from your participation will be kept strictly confidential, and we guarantee that it will not be sold or given to others or institutions without your permission," as long as the data is only shared/published in anonymous form.

As for the raw data file, you can either share the original file, as suggested by the reviewer, or state that you will share the data file used as input for the analyses (i.e., modified .xls, .txt, or .csv files).

Sincerely,

Francesca Peressotti

·

Basic reporting

The changes in response to my comments on the previous version are appropriate. (I did not check this as extensively for the other reviewer’s comments.) I appreciate the clearer characterization of oral reading and predictive language processing in general. Overall, this results in a more coherent and informative manuscript. It would still benefit from a thorough proofreading round, however, as I encountered frequent formulation issues.

Some rather domain-specific formulation issues that general proofreading might not catch:

l 64: ‘language beginners’- > beginning readers
l 73-74: ‘an explanation for the mechanism of prediction’ does not make sense
l 86: ‘engaging the production’ -> engaging the production system
l 87: ‘a final word’ -> the final word
l 149: ‘interactions between reading mode and the word predictability effect’ -> interactions between reading mode and word predictability effect’
l 172: ‘they will get’ -> they received
l 193-4: ‘the cloze value (i.e., predictability) between high and low predictable words was significant’ does not make sense
l 396: ‘speed of onset (SOA)’: description does not match acronym
l 396-7: ‘the level predicted’ does not make sense
l 419: ‘even though the existing studies’: sentence ends abruptly

Experimental design

In my previous review, I asked if compliance was monitored for the reading aloud task, i.e., did participants actually read aloud and did they do so correctly? I still think this is relevant to know, but the revised manuscript still does not provide that information.

I appreciate the additional analyses that were run using reading speed, even though these would have seemed more informative to me if they had modeled reading speed as a continuous variable rather than a dichotomous grouping variable. Also, in interpreting these analyses, it would be prudent to emphasize that they were conceived post-hoc and are potentially underpowered to detect complex interactions.

Validity of the findings

Parts of the discussion strike me as contradictory. In l 358-360 the authors write that “This implies that our findings lend support to the Prediction-by-Production hypothesis while not corroborating the Prediction-by-Association hypothesis.” while in l 367-368 they write that “An alternative explanation for the observed interactive effect on early eye movement measures is that the predictive processing operates analogously to the priming effect.” My understanding of prediction-by-association, as defined by Pickering and Gambi (2018), is that it is essentially semantic or associative priming. So, are the findings consistent with a semantic priming/prediction-by-association account or not?

The paragraph at l 425-434 contains only very broad speculation, without any references to existing work. I agree that these factors might play a role, but mentioning them is also a bit gratuitous, especially if no further motivation is given as to why each of these might matter and why it is important to investigate them further. My suggestion would be to either provide more specific motivation or to remove this paragraph.

·

Basic reporting

Thank you for considering my comments from the first round of review. Overall, I think the paper has improved, but another major revision is necessary. The main shortcomings of the current draft include: (1) the introduction could use some further clarification, particularly with regards to the different reading measures you used and the associated hypotheses (see comments below), (2) there are missing details in methods; please also ensure that you sorted out the informed consent issues; (3) I noticed mistakes in reporting the results in the paper (wrong numbers), it is essential these are corrected; (4) interpretation and discussion, please expand the discussion of null on skipping rates, please discuss the possibility that your finding is confounded by a slow-down in reading aloud, please be sufficiently careful when interpreting findings on late measures where you did not have clear apriori predictions, please rework the context that you have added to make sure your logic is clear and easy to follow, (5) adjust the abstract accordingly (e.g., interactions for some first pass measures; if you want to keep the claim that you show prediction-by-production impacts specifically lexical access and semantic integration, please make sure it is clear in your paper which measure is assictaed with what).

I would like to reiterate that I find the data contributed by this paper important, but more work is needed to allow peers to understand what was done here.

Literature review and study motivation

51-54 (line numbers based on revised draft with tracked changes on)
I feel this sentence is missing words – did you want to say that the role of prediction matters particularly in reading? In any case, you seem to claim that predition is important in reading because there are fewer cues – but it could be that prediction is less important there for the same exact reason, no? Maybe a safer argument would be to simply say that we have lots of evidence showing that prediction is engaged in reading, therefore it makes sense that you focused on reading too (you seem to imply this by discussing reading research in the following paragraphs). Or make any other argument that convincingly says it is important to test for prediction in reading.

72-77
This passage seems a bit out of place – what’s the purpose of talking about the brain if your study does not involve neuro- measures?

88-89
Could you unpack your logic here. Did you want to say reading aloud is important in certain populations, so it might be useful to understand which mechanisms it entails, e.g., prediction? As is, I am not sure if understand your argument.

90-91
The first sentence of this para feels unrelated to what preceds and follows.

105-106
the study of Lelonkiewicz et al. 2021 was not correlational, it was a behavioural experiment, please rephrase

127-128
The manipulation of Martin et al was not whether the production system was generally engaged, it was whether it was occupied with a concurrent production task

129-131
Martin et al offer the first direct experimental evidence, but it is not the first robust evidence -- many previous correlational or otherwise indirect studies were robust

197-204
For the sake of making your work accessible to your (overworked) peers, please revise this paragraph – it is very difficult to understand what argument you are making.
Moreover, in the previous draft you did not include any predictions for late measures of predictability. Also, in the current draft you only mention that late measures are ‘often reported’ (181-183). I assume then that you did not have clear predictions for prediction-by-production effects here? If so, please do not come up with predictions post-hoc, instead make it clear that you were exploring things wrt these variables. This should not prevent your work from being published imo. Later in the results and discussion you are interpreting findings on late measures – this is fine and offers some potentially interesting insights.

Raw data
In this round of review the authors also did not provide the raw files. Once again, in my opinion this is acceptable, but I flag this according to the journal’s policy. I appreciate the response provided by the authors, but I disagree -- EDF eye-tracker files are raw files, anything else is processed. By the way, there are R libraries for exporting edf as csv files, e.g., https://cran.r-project.org/web/packages/eyelinker/vignettes/basics.html

Table 1 – for the low predictability word, was it represent or delegates? You give two different translations of the original Chinese word used in your example sentence.

Table 2 – what are the values in brackets?

Figure 1 – in my previous review I argued that you should present data from *all* the measures you analysed. Please do so. In the new figure there seems to be no effect of predictability on FFD, GD, TRT in silent reading. Is this the case? IF so, please make it clear in the results/discussion, because this means you did not replicate the previous eye-tracking findings which, as you point out, mostly come from silent reading set-ups (Staub et al 2015).

Experimental design

On the issue of using 6 different predictability measures – in your reponse to my earlier comments you argued you wanted to capture parafoveal processing, early word processing, semantic integration. Please make this point in the introduction to your paper so it becomes clear to the readers, not only to the reviewers. Staub et al. 2015 offers a very good discussion of what aspects of processing might be captured by different reading measures. I would say my earlier worry about multiple comparisons (meaning inflating type I error due to looking for the effect on multiple measures simultaneously, without clear predictions which measures should vs should not be affected) stands as long as your predictions and the motivation for them are not made clear. But note that I do not ask you to come-up with predictions post-hoc; exploratory hypotheses are fine, even though theire results should be taken with caution, precisely due to the multiple comparison issue.

On the issue of analysis code – thank you for submitting updated datafiles and the additional readme explaining how you pre-processed data. Please post this readme in the supplement and refer the reader of the paper to it (methods). I have two comments about the readme: in 1, please add an English translation of the sentence; in 4, I noticed that trial counts add to 165 not 160? Please check this. In the future, consider using R to do pre-processing in an automatic – and easily reproducible – way.

Thank you for providing the R files with code for your analysis. Please make sure you describe the final models in the paper for each analysis that you report (more on this below). Please post your R files in a public repository or attach in supplement.

On the issue of ethical consent – I have previously noted that in the consent form participants agreed to the following: ‘All data and information collected from your participation will be kept strictly confidential, and we guarantee that it will not be sold or given to others or institutions without your permission.’ This effectively prohibits you from sharing any of their data. You replied that data ‘will not be disclosed in any identifiable manner’ – this is beyond the point, because you did not phrase it like that when collecting consent. You also stated that you have produced a second, updated consent form and collected signatures from all your participants on this new consent, now allowing you to share their data.

I did not have time to check this (are there documents confirming you have new signatures?), I leave it to the editorial team to confirm that you have successfully addressed this issue.

For the sake of participants privacy, I suggest you replace the participants labels constructed of bits of their names with a randomly assigned integer or letter string.

289-299
On my earlier point about posting the protocol from power analysis. I understand you used a web-based implementation of power analysis, but you still could share the protocol – how did you set the parameters? A screenshot would do, you could add it to the supplement.

301
Did you mean to say participants ‘received’ this reimbursement?

347-368
The purpose of my previous questions about the procedure was not to know it for myself (thank you for answering though), I meant to ask you to include these details in the paper. Please do so – comprehension questions, calibration, experiment duration.

378-379
Please cite papers using your exclusion procedure – as you did in your reponse to my previous comments

382
See my comment about number of excluded trials not adding up as per your readme file

Validity of the findings

As I outlined before, it would be great to have a clearer idea what (if any) predictions you had for each of the predictability measures. Where you did not make a prediction and you were exploring, please alert the reader to this fact.

Reporting results from the comprehension questions analysis – please report also the interaction effect which is critical here; it would be nice to have at least a sentence explaining why did you test for differences in comprehension.

Thank you for explaining the model structure to me in your reponse, but do explain it also to the readers - for which variables did you run LME vs GLMM, what was the structure of each model.

393
Please add results from log-transformed analyses to the supplement, so that the reader can judge the level of similarity for themselves. Add them to R files too.

397
Did you mean to say the full model results are reported in the supplement?

402
I see some mistakes in the results you report, for example the main effect of predictability as reported in the paper does not match the results from your model as reported in your R. Please double-check all results in the paper, this is very important.

399-410
Predictability effects – you conclude that you replicated the typical effects, but I am not sure. Most studies reporting these effects involve silent reading, but in your study the high vs low effect is driven by reading aloud for almost all measures. I think this is something that should be made clear already here. The conclusion that you replicate might not be entirely warranted.

435-450 and later in discussion
The three-way reading speed*mode*predictability analysis - to be honest, I do not understand what this is meant to accomplish. Please explain (in the paper) and add to your R files. To my mind, the predictability*reading mode interaction in your data might be due to the fact that reading aloud was associated with a sufficient slow down – I observed the same in Exp 1 of my 2021 study. If reading aloud was doing something general to prediction then you should also observe an interaction effect for skipping rates. Please address this in your discussion.

483-488
See my earlier comment, be careful when drawing conclusions about predictability effects. Also, do you refer only to reading in Chinese here? I noticed you happen to cite only Chinese studies. Please clarify or expand citations.

508-517
I can’t follow your logic here. You suggest that where and when of eye movements are independent and you use this to explain the null effect on skipping rates (where) vs significant effects for reading times (when). So what if these are independent? In past studies skipping rate was a reliable measure of predictability, and also in your study you report predictability effects with regards to this measure. Why then prediction-by-production would not impact skipping rates? Please expand your discussion of the null effect of skipping (bearing in mind null does not constitute evidence for absence).

519-521
I can’t follow this sentence, perhaps you are missing some words here? Please rephrase. Overall, I once again invite you to go through your paper with a fluent English speaker.

546-559
I am a fellow psychologist and I have published on prediction before and yet I do not understand the point you are making here, please rephrase.

---

## Round 0.3 · Minor Revisions

Dear authors,

As you can see from the attached comments, both Reviewers consider that your manuscript has really improved as, pending some minor revisions, it is valuable for publication in Peerj. I invite you to address the minor points raised by the two reviewers and resubmit the manuscript. I reserve the right to read it personally and potentially make a final decision regarding it.

·

Basic reporting

no comment

Experimental design

no comment

Validity of the findings

I have carefully read the revised manuscript and the response to the editor’s and reviewers’ comments. I appreciate the authors’ effort in following up on all of the comments. I have a few remaining issues:

1. The ANCOVA that the authors now report wasn’t exactly what I was suggesting. Rather, I was suggesting substituting a continuous variable for the binary variable ‘reading speed’ in the analyses that probed for three-way interactions between predictability, reading mode, and reading speed. This should not be a problem within the HLM framework the authors have adopted. Merely adding a covariate to ‘control’ for reading speed, without testing whether reading speed modulates the interaction between reading mode and predictability, is not very informative, as far as I am concerned.

Still, whether modeled as a binary variable or a continuous variable, these analyses do not rule out the possibility that the slower reading pace in the oral reading condition is causally involved in the stronger contextual predictability effect in that condition. Even if the effect were not graded with reading speed within either modality, participants still read more slowly in the oral reading condition than in the silent reading condition. The additional analyses cannot remove this confound. More research is needed to tease out why contextual predictability effects are larger in oral reading. I think this needs to be stated more clearly.

2. I agree with the point raised by the other reviewer about multiple comparisons, especially in view of the lack of clearly motivated a priori hypotheses for each dependent measure. I think this point has not yet been sufficiently addressed. While the use of multiple measures of early and late processing is quite common, the issue remains that when you report results for six dependent variables, the probability of at least one of these being a false positive increases proportionally. I think it would be best if the authors use a standard procedure for correcting their alpha.

Additional comments

no comment

·

Basic reporting

I thank the authors for accommodating most of my comments. In my opinion the paper has improved greatly since the first submission and is now sufficiently mature for publication, pending my minor comments below. I thank the authors for their patience and collaboration in the review process!

In the previous review round, I suggested the abstract should be updated to reflect the changes made during the review process. The authors chose to ignore this, but I do believe in its current form the abstract is misrepresenting the contribution of the paper. Specifically, in Background, rephrase the statement: 'However, this hypothesis lacks empirical evidence from sentence reading contexts.' Given the studies of Martin et al. and Lelonkiewicz et al. have used sentence contexts to demonstrate prediction-by-production, this statement is incorrect. In Results, 'interactive effects were observed in first-pass reading measures', you did not find effects in all measures, you could simply add *some*.

Likewise, in Discussion lines 341-343 please add *some* or *several* when claiming interactions on your reading measures.

I also suggest caution when claiming to replicate the 'typical' predictability effects observed in studies that involved silent reading only -- on almost all measures in the present study high vs low comparison was not significant in the silent condition. Thus, please adjust the section title in line 279.

In the section Result --> Results

Experimental design

-

Validity of the findings

-

Additional comments

Table 2 -- if you report all results regarding the comparison of low vs high sentences in text then you can do away with Table 2. This is just a suggestion.

---

## Round 0.4 · accepted · Accept

It is my pleasure to accept your manuscript in its current form for publication in PeerJ.